Comparative transcriptomic analysis reveals potential mechanisms for high tolerance to submergence in arbor willows

Chen Yanhong 1 2
Yang Jie 1 2
Guo Hongyi 1 2
Du Yawen 1 2
Liu Guoyuan 1 2
Yu Chunmei 1 2
Zhong Fei 1 2
Lian Bolin 1 2
Zhang Jian 1 2 ntdxylzw@163.com
1 School of Life Science, Nantong University , Nantong, Jiangsu Province , China
2 Key Lab of Landscape Plant Genetics and Breeding , Nantong , China
Sun Genlou
Electronic publication date: 2022 Feb 3
Publication date: 2022
Volume: 10
Electronic Location ID: e12881
Received 2021 Aug 5; Accepted 2022 Jan 13
Copyright: © 2022 Chen et al.
Copyright year: 2022
Copyright holder: Chen et al.
License: This is an open access article distributed under the terms of the Creative Commons Attribution License, which permits unrestricted use, distribution, reproduction and adaptation in any medium and for any purpose provided that it is properly attributed. For attribution, the original author(s), title, publication source (PeerJ) and either DOI or URL of the article must be cited.
License URL: https://creativecommons.org/licenses/by/4.0/

Keywords: Submergence stress, Hypoxia, Salix, Transcriptomic analysis, Hormone, Transcription factors, Anaerobic respiration pathway

Funding: National Natural Science Foundation of China 31971681 Technology Innovation Center for Land Spatial Eco-restoration in Metropolitan Area, Ministry of Natural Resources and the Fundamental Research Funds for the Central Universities CXZX2021A03 Basic Science Research Project of Nantong City JC2020157 This work was supported by the National Natural Science Foundation of China (No. 31971681) and the Technology Innovation Center for Land Spatial Eco-restoration in Metropolitan Area, Ministry of Natural Resources and the Fundamental Research Funds for the Central Universities (grant no. CXZX2021A03), and by the Basic Science Research Project of Nantong City (grant no. JC2020157). The funders had no role in study design, data collection and analysis, decision to publish, or preparation of the manuscript.

==============================
Background

Submergence threatens plant growth and survival by decreasing or eliminating oxygen supply. Uncovering the complex regulatory network underlying the tolerance of Salix to submergence and identifying the key regulators are important for molecular-assisted breeding of Salix.

Methods

In this study, we screened germplasm resources of arbor willows and discovered both submergence-tolerant and submergence-sensitive varieties. Then, by performing RNA-seq, we compared the differences between the transcriptomes of two varieties, i.e., the submergence-tolerant variety “Suliu 795” and the submergence-sensitive variety “Yanliu No. 1,” and the different submergence treatment time points to identify the potential mechanisms of submergence in Salix and the unique approaches by which the variety “Suliu 795” possessed a higher tolerance compared to “Yanliu No. 1”.

Results

A total of 22,790 differentially expressed genes were identified from 25 comparisons. Using gene ontology annotation and pathway enrichment analysis, the expression pattern of transcriptional factors, important players in hormone signaling, carbohydrate metabolism, and the anaerobic respiration pathway were found to differ significantly between the two varieties. The principal component analysis and qRT-PCR results verified the reliability of the RNA sequencing data. The results of further analysis indicated that “Suliu 795” had higher submergence tolerant activity than “Yanliu No. 1” because of three characteristics: (1) high sensitivity to the probable low oxygen stress and initiation of appropriate responding mechanisms in advance; (2) maintenance of energy homeostasis to prevent energy depletion under hypoxic stress; and (3) keep “quiescence” through fine-tuning the equilibrium between phytohormones GA, SA and ethylene.

Introduction

Water plays a central role in the interaction between plants and the environment. However, excess water can result in flooding, which negatively affects the developmental periods of plant lifecycle, from seed germination to vegetative and reproductive growth (Zhou et al., 2020). Flooding can be classified as waterlogging or submergence (Fukao et al., 2019). Waterlogging occurs when the roots of plants are submerged in water and the soil is saturated, while submergence occurs when the aerial plant tissues are fully submerged in water (Sasidharan et al., 2017). Water surrounding roots or whole plants disrupts the exchange of oxygen between the plant and the atmosphere, leading to low oxygen availability (hypoxia <21% O2) or even complete absence of oxygen (anoxia) in plant tissue, which is detrimental to plant survival (Loreti, van Veen & Perata, 2016). Wetland plant species such as Juncus effuses have developed constitutive aerenchyma anatomical structures to facilitate gas diffusion between roots and the aerial environment (Fukao & Bailey-Serres, 2008). In crops such as maize, aerenchyma development can be induced by flooding or hypoxia (Visser & Bögemann, 2006). Another study reported a flood-adapted structure/organ that improves oxygen supply in plants during flooding via adventitious roots (Sauter, 2013). During flooding or submergence, flooding-tolerant herbaceous plants and trees produce shoot-borne adventitious roots, as seen in the perennial wetland species Cotula coronopifolia and Meionectes brownie (Calvo-Polanco, Señorans & Zwiazek, 2012; Rich, Ludwig & Colmer, 2012; Sauter, 2013). Adventitious roots replace the sediment root system to improve plant growth by shortening the distance between the source of oxygen supply and oxygen consumer (cells, tissues, and organs). In addition, adventitious roots can reduce radial oxygen loss because of poorly developed endodermis or produce oxygen in situ (Calvo-Polanco, Señorans & Zwiazek, 2012; Rich, Ludwig & Colmer, 2012; Sauter, 2013).

A complex regulatory network in plant submergence or waterlogging tolerance has been revealed in model plants, such as Arabidopsis and rice, in which metabolic regulation pathways and phytohormone signaling pathways are interconnected (Fukao et al., 2006; Xu et al., 2006; Hattori et al., 2009; Licausi et al., 2010; Niroula et al., 2012; Voesenek & Bailey-Serres, 2015; Fukao et al., 2019). Upon submergence or waterlogging stress, the depletion of oxygen triggers the anaerobic respiration metabolism pathway (Voesenek & Bailey-Serres, 2015; Fukao et al., 2019). Sugar metabolism enzymes, such as alcohol dehydrogenase (ADH), pyruvate decarboxylase (PDC), and sucrose synthase (SUS) are significantly upregulated during waterlogging (Ismond et al., 2003; Kato-Noguchi & Morokuma, 2007). Gibberillic acids (GA), ethylene, auxin, and ABA are major phytohormones involved in submergence or waterlogging tolerance by promoting the formation of adventitious roots and aerenchyma and regulating the secondary metabolism pathway or plant growth, such as shoot elongation (Jackson, 2008; Voesenek & Bailey-Serres, 2015; Qi et al., 2020). Flood-tolerant plants have developed many survival strategies, including development of adventitious roots and aerenchyma, the “escape” (low-O2 escape syndrome, LOES) and “quiescence” (low-O2 quiescence syndrome, LOQS) responses (Fukao & Bailey-Serres, 2008; Hattori et al., 2009; Akman et al., 2012; Niroula et al., 2012; Voesenek & Bailey-Serres, 2015). LOES and LOQS are two extreme strategies. The key regulators in “escape” or “quiescence” are the evolutionarily conserved group VII ethylene response factor (ERF-VII) transcription factors (TFs), which have been extensively investigated in rice and Arabidopsis. One study found that the expression of SNORKEL1 and 2 (SK1/2) was induced by the accumulation of ethylene due to submergence in deep water rice. SK1/2 activated the expression of the GA synthesis gene GA-20 oxidase (GA20ox), which promotes internode elongation and helps plants evade hypoxia stress (Hattori et al., 2009; Ayano et al., 2014). Another ERF-VII member from lowland rice, Submergence1 (Sub1A), was found to exert the antithetical function of SK1/2, which maintains “quiescence” during flooding (Xu et al., 2006; Fukao & Bailey-Serres, 2008; Niroula et al., 2012). By enhancing the mRNA levels of Slender Rice-1 (SLR1)/SLR1 Like-1 (SLRL1) and GA catabolic gene GA2ox7, Sub1A was found to inhibit GA signaling and production. Sub1A can also restrict the rate and extent of catabolism of starch and soluble sugars (Fukao & Bailey-Serres, 2008; Barding et al., 2012; Niroula et al., 2012). The five group VII ethylene-responsive TFs (ERF-VIIs) of A. thaliana, including RELATED TO APETALA2.12 (RAP2.12, AtERF75), RAP2.2 (AtERF74), RAP2.3 (AtERF72), and hypoxia responsive ERFs (HRE1/AtERF73, HRE2/AtERF71), mainly participate in the transcription of anaerobic metabolism genes to construct metabolic acclimations (Licausi et al., 2010; Gibbs et al., 2011; Licausi et al., 2011). The oxygen levels under hypoxic stress conditions influence the localization and stabilities of ERF-VIIs proteins. The stability of ERF-VII proteins is modulated by the presence of molecular oxygen via the N-end rule protein degradation pathway (Hinz et al., 2010; Licausi et al., 2010; Papdi et al., 2015; Prabhakar & Semenza, 2015). Trihelix TF hypoxia response attenuator 1 (HRA1) directly binds RAP2.12, which plays an important role in the N-end rule pathway (Giuntoli et al., 2014). In addition to ERF-VII and Trihelix TF, the study of angiosperm flooding response using high-throughput sequencing identified three other significantly enriched TF motifs in submergence–upregulated family genes, namely a basic helix-loop-helix (bHLH), MYB, and WRKY-type motif. This research indicated the important roles of bHLH, MYB, and WRKY in the flooding response (Reynoso et al., 2019).

Several studies have reported the regulation mechanism of flooding tolerance in species other than rice and Arabidopsis. These researches focused on the ERF-VIIs genes too (Fan et al., 2018; Pan et al., 2019; Yu et al., 2019; Luan et al., 2020). For example, the overexpression of ZmEREB180, a maize ERF-VII member in Arabidopsis and maize, has been found to enhance the survival rate of plants after long-term waterlogging stress through the enhanced formation of adventitious roots (ARs) (Yu et al., 2019). Phenotypic and physiological analyses of transgenic Arabidopsis overexpressing barley HvERF2.11 indicated that HvERF2.11 works as a positive regulator in plant waterlogging tolerance by improving antioxidant and ADH enzyme activities (Luan et al., 2020). The overexpression of AdRAP2.3 in transgenic tobacco also improved the expression levels of waterlogging marker genes, such as PDC and ADH (Pan et al., 2019). Other studies on flooding tolerance mechanisms in non-model plants are mainly carried on by RNA sequencing, such as with Chrysanthemum morifolium, maize, cucumber, Zombi pea (Vigna vexillata), and Brassica napus (Zou et al., 2013; Arora et al., 2017; Xu et al., 2017; Zhao et al., 2018; Butsayawarapat et al., 2019). These transcriptomic studies have reported similar differentially expressed genes (DEGs) that function as TFs, regulators, or factors in hormone synthesis and signaling, ROS-scavenging, N-end turnover, anaerobic respiration, and carbohydrate metabolism. A number of studies have identified the mechanisms underlying the differences in waterlogging tolerance in different varieties, including Chrysanthemum morifolium and Zombi pea (Zhao et al., 2018; Butsayawarapat et al., 2019).

The molecular mechanism of flooding tolerance was uncovered by studies primarily on herbaceous plants, particularly rice and Arabidopsis. With regards to the mechanisms for flooding tolerance in trees, studies using transcriptome, physiological, and metabolome analyses have identified few key regulator genes, despite the enormous economic, ecological, and social importance of forest ecosystems (Juntawong et al., 2014; Kreuzwieser & Rennenberg, 2014; Qi et al., 2014; Zhang et al., 2017; Salvatierra et al., 2020). In general, trees have been found to survive O2 deficiency stress by slowing down the anabolic processes and maintaining a steady supply (Kreuzwieser & Rennenberg, 2014; Salvatierra et al., 2020). Transcriptomic analysis on “Zhongshansa,” which shows a remarkable tolerance to waterlogging, showed that ROS detoxification and energy maintenance were the primary coping mechanisms of this species.

Salix, such as the arbor willow, is a tree species that provides significant environmental protection within its ecosystems and has a strong tolerance to flooding stress (Kuzovkina & Volk, 2009). The characterization of responses of willows to partial flooding has mainly focused on physiological aspects (Li et al., 2004; Rodríguez et al., 2018), and little is known about the responses of these trees to submergence, as well as the molecular mechanisms in submergence response. Therefore, to elucidate the genetic basis of the submergence tolerance of Salix and identify its key regulators, we analyzed the transcriptomes of roots exposed to submergence from two Salix varieties with different submergence tolerance abilities and identified several hub genes in hypoxia tolerance. Based on our findings, we propose a potential mechanism by which submergence-tolerant varieties obtain high tolerance abilities upon submergence stress.

Materials and Methods

Screening submergence-tolerant and submergence-sensitive varieties in arbor willow germplasm

One-year-old stem cuttings (length, 8–10 cm; coarse, 2–3 mm) of 13 arbor willow varieties (“Zhuliu,” “Suliu 172,” “Suliu 795,” “Suliu 932,” “Longzhualiu,” “Dongying No. 1,” “J1010,” “Ziganliu,” “J2187,” “J2087,” “Bohailiu No. 1,” “Yanliu No. 1,” and “287”) were immersed in water to mimic submergence stress. Three submergence stress treatment repeats and a control experiment were set for each willow variety, with 15 stem cuttings per treatment. The emergence time and number of buds and roots from the stem cuttings were counted every 3 days to determine the tolerant and sensitive varieties. The duration of submergence stress treatment was 66 days. During the submergence stress period, varieties with more buds/roots and longer buds/roots were characterized as tolerant to submergence. Total chlorophyll content and malondialdehyde (MDA) contents were measured using kits according to the manufacturer’s instructions (Solarbio, Beijing, China).

Treatment and collection of samples for RNA sequencing

Stem cuttings (length, 8–10 cm; coarse, 2–3 mm) of submergence-tolerant and submergence-sensitive willow varieties were cultured in Hoagland liquid medium for 20 d to induce the emergence of roots and buds. The liquid medium surface was 4 cm above the end of the stem cuttings. The stem cuttings with newly emerged roots and buds were completely immersed in water, and root samples were collected after 0, 4, 12, 24, and 48 h of submergence stress for RNA sequencing. The samples were labeled as WR (waterlogging resistance)-CK, WR-4h, WR-12h, WR-24h, WR-48h, and WSYL (waterlogging sensitive Yanliu)-CK, WSYL-4h, WSYL-12h, WSYL-24h, and WSYL-48h for the five samples from the submergence-tolerant (Suliu 795) and submergence-sensitive (Yanliu No. 1) varieties, respectively. WR-CK sample means submergence-tolerant variety roots not treated with submergence stress; WSYL-CK sample means submergence-sensitive variety roots not treated with submergence stress. The experiments were repeated in triplicate, and a total of 30 samples were collected for RNA sequencing.

RNA sequencing, data processing, and gene annotation

Total RNA from all samples was extracted and purified using the same protocol and reagent as reported by Chen et al. (2020b). cDNA library construction was performed according to the protocol described by Chen et al. (2020b). After completing library construction, Qubit2.0 and Agilent 2100 were used to detect library concentration and library insert size; Q-PCR was used to accurately measure library concentrations for library quantification. After passing library quality inspection, the library preparations were sequenced on an Illumina platform, and paired-end reads were generated. Clean data (clean reads) were obtained from the screening of raw data (raw reads). The Q20, Q30, GC-content, and sequence duplication levels of the clean data were calculated to guarantee that all downstream analyses were based on clean data with high quality. Principal component analysis (PCA) was performed to verify the reliability of the repeated experiments.

These clean reads were then mapped to the Salix matsudana reference genome sequence V1.0 (Zhang et al., 2020). TopHat2 software was used to map the reference genome (Kim et al., 2013). Using BLAST+ (v2.2.23) software (http://blast.ncbi.nlm.nih.gov/Blast.cgi) with default parameters, gene function was annotated based on the seven databases routinely used in gene annotation (Qi et al., 2014; Zhao et al., 2018).

Differential expression analysis and GO and KEGG pathway enrichment analysis on differentially expressed genes

The gene expression levels were estimated by fragments per kilobase of transcript per million fragments mapped (Jin, Wan & Liu, 2017). Differential expression analysis of two conditions/groups was performed using DESeq2 (Wang et al., 2009). False discovery rate was controlled through Benjamini and Hochberg’s approach by adjusting the resulting P value. Genes both with an adjusted P value < 0.01, and fold change value > 2 were confirmed as DEGs. Gene Ontology (GO) enrichment analysis of DEGs was carried out by the GOseq R packages (Young et al., 2010). The statistical enrichment of DEGs in the KEGG pathways was analyzed using KOBAS software (Mao et al., 2005).

Quantitative real-time PCR

The expression patterns of selected candidate DEGs were verified using quantitative real-time PCR to test the reliability of RNA sequencing. Ten RNA samples for RNA sequencing were reverse transcribed, and qRT-PCR was carried out, as previously reported (Zhang et al., 2020). The actin gene (SapurV1A.0655s0050.1) in Salix was used as the reference gene. The gene-specific primers for the 15 selected genes are listed in Table S1. Relative expression was measured using the 2−ΔΔCt method. The expression level of “WR-CK” was selected as the reference and normalized to a relative expression of 1.0.

Co-expression gene network construction of DEGs and hub gene identification

DEGs with FPKM (≥0.5) were used to construct the co-expression gene network by R package weighted gene correlation network analysis (WGCNA) (Langfelder & Horvath, 2008). The minimum module size and the minimum height for merging modules were set to 30 and 0.2, respectively. The graphic networks on modules were visualized using Cytoscape 3.7.0 software (Shannon et al., 2003). GO and KEGG pathway enrichment analyses were performed on DEGs in modules with high correlation coefficients and corresponding lower P-values. Genes in the modules with the highest connectivity values were identified as hub genes.

Expression pattern analysis of DEGs involved in hormone and metabolism pathways related to submergence stress response

Using BLAST, we identified the S. matsudana homolog genes of key Arabidopsis metabolism and signaling pathways in flooding response, including carbohydrate metabolism, ROS signaling pathway, anaerobic respiration, N-end rule pathway, phytohormone (ethylene, GA, and salicylic acid (SA)) synthesis, and signaling pathways. DEGs from these genes were identified and their expression patterns were illustrated using the heatmap illustrator of TBtools (Chen et al., 2020a). Transcription factor prediction analysis was performed using the BMKCloud platform (www.biocloud.net). The statistical analysis of total TF genes and DEGs of TF genes was carried out based on the analysis. The expression patterns of the putative DEGs of the TFs in the RNA-seq were presented by heatmaps using TBtools (Chen et al., 2020a).

Results

“Suliu 795” and “Yanliu No. 1” were selected as submergence-tolerant and submergence-sensitive representative varieties, respectively

Thirteen varieties, namely “Zhuliu,” “Suliu 172,” “Suliu 795,” “Suliu 932,” “Longzhualiu,” “Dongying No. 1,” “J1010,” “Ziganliu,” “J2187,” “J2087,” “Bohailiu No. 1,” “Yanliu No. 1,” and “287,” were selected for submergence stress experiments.

The total root number, total root length, total shoot number, and total shoot length of 13 varieties under different submergence treatment times are summarized in Fig. 1 and Table S2. In the total root number panel, the top four varieties were “Zhuliu,” “Suliu 172,” “Suliu 795,” and “Suliu 932” (Fig. 1A). In the total root length panel, the top four varieties were “Zhuliu,” “Suliu 172,” “Suliu 932,” and “Suliu 795” (Fig. 1B). In the total shoot number panel, the top four varieties were “Longzhualiu,” “Dongying No. 1,” “Suliu 795,” and “287” (Fig. 1C). In the total shoot length panel, the top four varieties were “Suliu 795,” “Longzhualiu,” “Zhuliu,” and “Dongying No. 1” (Fig. 1D). These seven varieties had the ability to maintain root and bud growth under long-term flooding submergence. Some showed advantages in root growth (“Zhuliu,” “Suliu 172,” and “Suliu 932”), others advantages in bud growth (“Longzhualiu,” “Dongying No. 1”), and only “Suliu 795” predominated in all four evaluation indicators. In addition to these seven varieties, another six varieties (“J1010,” “Yanliu No. 1,” “J2187,” “J2087,” “Ziganliu,” and “Bohailiu No. 1”) stopped growing or maintained a slow growth rate under long-term submergence. We also made analysis on two physiological indicators including total Chlorophyll content and MDA content. From the results of MDA content, we found that the MDA content in ‘Suliu 795’ is the lowest in all samples detected (Fig. S1A). After submergence stress, the total chlorophyll content in all samples decreased, but the smallest reduction range was found in the ‘Suliu 795’ (Fig. S1B). Since “Yanliu No. 1” is a variety of Salix found in salty land, we selected “Suliu 795” and “Yanliu No. 1” as representatives for submergence-tolerant and submergence-sensitive varieties, respectively. The growth phenotype under normal conditions (hydroponics culture) and submergence stress of “Suliu 795” and “Yanliu No. 1” are presented in Figs. 1E and 1F.

Figure 1 Four growth indicators of the arbor willow germplasm under submergence stress, and the phenotypes of the submergence-tolerant and submergence-sensitive varieties.

Thirteen varieties selected for the submergence stress experiments and four indicators, including total root number, total root length, shoot number, and shoot length, are listed: (A) total root number; (B) total root length; (C) total shoot number; (D) total shoot length; (E) growth phenotype under control condition (hydroponics culture) and submergence stress of “Suliu 795” (66 days); (F) growth phenotype under control condition (hydroponics culture) and submergence stress of “Yanliu No. 1” (66 days) (F).

RNA sequencing and data quality control

To identify the differences in the mechanisms of the response to submergence between the sensitive and tolerant cultivars, 30 libraries (three repeat libraries for one sample) from the root tissues of the varieties, i.e., WR-CK (“Suliu 795” control), WR-4h, WR-12h, WR-24h, WR-48h, WSYL-CK (“Yanliu No. 1” control), WSYL-4h, WSYL-12h, WSYL-24h, and WSYL-48h, were constructed in this study. After controlling for the quality of the sequencing data, 192.46 Gb clean data were obtained using Illumina HiSeq. The minimum Q30 and average GC contents were 91.55% and 44.23%, respectively (Table S3). From the alignment results using the S. matsudana genome as the reference genome, the alignment ratio of samples ranged from 71.52% to 85.99%, of which approximately 90% reads were mapped to the exon region (Table S3).

An assessment of the relationships between biological replicates is essential for analyzing transcriptome sequencing data for a project with biological replicates. Thus, PCA was performed to confirm the uniformity between biological replicates of the 10 group samples (Fig. 2A). Three biological replicates from 10 groups were clustered tightly and separated distinctly from each other, indicating the reliability of our RNA-seq results.

Figure 2 Statistical analysis on RNA sequencing results.

(A) PCA analysis. (B) Statistical analysis of differentially expressed genes (DEGs) from 25 comparisons (5 categories). (C) Venn diagrams of upregulated and downregulated DEGs from the comparisons of categories A, B, and C.

Comparative transcriptomic analyses and GO annotation and pathway enrichment analysis on DEGs

To identify the differences in the expression profiles between the two varieties and the dynamic changes following the extension of submergence time, 25 comparisons were made to identify DEGs between different treatment time points in the same variety and between the two varieties (Tables S4–S8), 22790 DEGs were identified from 25 comparisons (Table S9), the sequences of all DEGs were listed in Files S1 and S2. The 25 comparisons fell into five categories: category A included 4 comparisons between different treatment time points and the control in the submergence-tolerant variety “Suliu 795” (WR-CK/WR-4h, WR-CK/WR-12h, WR-CK/WR-24h, and WR-CK/WR-48h); category B included 4 comparisons between different treatment time points and the control in the submergence-sensitive variety “Yanliu No. 1” (WSYL-CK/WSYL-4h, WSYL-CK/WSYL-12h, WSYL-CK/WSYL-24h, and WSYL-CK/WSYL-48h); category C included 5 comparisons between two varieties at 5 treatment time points (WR-CK/WSYL-CK, WR-4h/WSYL-4h, WR-12h/WSYL-12h, WR-24h/WSYL-24h, and WR-48h/WSYL-48h); category D (WR-4h/WR-12h, WR-4h/WR-24h, WR-4h/WR-48h, WR-12h/WR-24h, WR-12h/WR-48h, and WR-24h/WR-48h) and category E (WSYL-4h/WSYL-12h, WSYL-4h/WSYL-24h, WSYL-4h/WSYL-48h, WSYL-12h/WSYL-24h, WSYL-12h/WSYL-48h, and WSYL-24h/WSYL-48h) represent comparisons between different treatment time points in “Suliu 795” and “Yanliu No. 1”, respectively. The upregulated and downregulated DEGs in each comparison are shown in Fig. 2B. In category A and B, more upregulated DEGs were only found in two comparisons between the CK and 4 h treatments, while in the other six comparisons, the number of downregulated DEGs was higher than that of upregulated DEGs. These data indicate that upon longer submergence, the plant adapted to low oxygen levels by slowing down cellular activity. Except for the comparisons between the CK and 4 h treatments, the tolerant variety “Suliu 795” had more DEGs than the sensitive variety “Yanliu No. 1” during longer submergence stages (Fig. 2B, Tables S4–S8), demonstrating that “Suliu 795” had a stronger response to submergence than “Yanliu No. 1” at the transcription level. The highest number of DEGs was observed in the WR-CK/WR-24h comparison, with 3,383 upregulated DEGs and 5,941 downregulated DEGs. In five comparisons in category C, an average of 5,000 DEGs were found, and the WR-CK/WSYL-CK comparison had 4,820 DEGs, which suggested that there were significant differences in gene expression between the two cultivars during hydroponic rooting and response to submergence stress. In categories D and E, the smaller number of DEGs in comparisons WR-24h/WR-48h (a total of 1,259 DEGs, including 747 up- and 512 downregulated genes) and WSYL-24h/WSYL-48h (a total of 490 DEGs including 220 up- and 270 downregulated genes) indicated a similar expression pattern at the 24 h and 48 h treatment time points in both varieties.

Venn diagrams were drawn to reveal overlapping DEGs in the different categories. In category A, there were 1,414 overlapped DEGs, with more downregulated DEGs (800) than upregulated DEGs, whereas in sensitive variety category B the number of overlapped DEGs was 2,180, and that of the upregulated DEGs was more than twice as many as that of the downregulated DEGs (Fig. 2C). The percentages of overlapping DEGs in categories A and B were 5.19% and 9.16%, respectively. In five comparisons of category C, there were a total of 1,259 overlapped DEGs (650 upregulated DEGs, 609 downregulated DEGs), accounting for only 5% of the total DEGs (Fig. 2C). The GO annotations of all overlapped DEGs were listed in six sheets of Table S10. From the table, we find that the top terms in biological processes (BP) and molecular functions (MF) were same in all overlapped DEGs groups, which are metabolic process (BP, GO:0008152) and binding (MF, GO:0003824).

To identify the major functional categories represented by the DEGs in different comparisons, gene ontology (GO) enrichment analysis, including the analysis of biological processes (BP), molecular functions (MF), and cellular components (CC), was performed. The results of BP enrichment terms (top 20 according to P-value) from upregulated DEGs and downregulated DEGs in categories A and B are illustrated in Fig. 3 and Tables S11–S12. The BP term “oxidation-reduction process (GO:0055114)” was found in all columns, indicating radical changes in the regulation of energy metabolism under submergence stress (Fig. 3). In the tolerant variety, after 4 h submergence treatment, the terms “negative regulation of catalytic activity (GO:0043086),” “negative regulation of endopeptidase activity (GO:0010951),” “negative regulation of nucleic acid-template transcription (GO:1903507),” and “regulation of protein serine/threonine phosphatase activity (GO:0080163)” were enriched in upregulated DEGs, indicating that plants slow down cellular activity upon stress (Fig. 3). The term “secondary metabolite biosynthetic process (GO:0044550)” was enriched in both upregulated DEGs and downregulated DEGs in WR-CK/WR-4h comparison, while after 8 h under stress conditions, this term was only listed in downregulated DEGs (Fig. 3). After 12 h of stress, new terms including “carbohydrate metabolic process (GO:0005975),” “nucleoside metabolic process (GO:0009116),” “response to nitrate (GO:0010167),” “nitrate transport (GO:0015706),” and “starch metabolic process (GO:0005982)” were enriched in upregulated DEGs. In comparison, in WR-CK/WR-24h and WR-CK/WR-48h the term “anaerobic respiration (GO:0009061)” was enriched in upregulated DEGs; the term “detection of hypoxia (GO:0070483)” was also overrepresented in upregulated DEGs of WR-CK/WR-48h comparison (Fig. 3). Some photosynthesis-related terms were found to be enriched in the columns from the upregulated regulated DEGs of category A. The term “photosynthesis, light harvesting in photosystem I (GO:0009768)” was enriched in three comparisons, including WR-CK/WR-12h, WR-CK/WR-24h, and WR-CK/WR-48h. Three terms, including “photosynthesis, light harvesting in photosystem I (GO:0009768),” “photosynthesis (GO:0015979),” and “chlorophyll biosynthetic process (GO:0015995),” were enriched in upregulated DEGs of WR-CK/WR-48h comparison (Fig. 3). In the submergence-sensitive variety, several terms were identical, such as “anaerobic respiration (GO:0009061),” while others were different. Photosynthesis-related terms were only found in the downregulated DEGs of WYSL-CK/WYSL-4h and WYSL-CK/WYSL-12h comparisons. The term “regulation of transcription, DNA-templated (GO:0006355)” was not listed in category A, but was enriched in upregulated DEGs in all comparisons from category B. The terms “hydrogen peroxide catabolic process (GO:0042744)” and “cellular oxidant detoxification (GO:0098869)” were enriched in the upregulated DEGs of WYSL-CK/WYSL-48h but not in those of WR-CK/WR-48h (Fig. 3).

Figure 3 Biological processes in gene ontology (GO) enrichment analysis of DEGs during submergence stress in the submergence-tolerant and submergence-sensitive varieties of Salix.

GO enrichment analysis was performed using Blast2GO. Only significantly enriched terms (top 20) with corrected P < 0.05 were indicated. The color and size of each point represents the ‒log10 (FDR) values and enrichment scores. A higher ‒log10 (FDR) value and enrichment score indicate a greater degree of enrichment. Red and blue labeled-terms represent the different and identical terms in different comparisons, respectively.

Large number of transcription factors is involved in the submergence stress response in Salix

Using BLAST, 4,584 TF genes were identified from 62,400 annotated genes, 1,844 TF genes were DEGs, and the top five TF family categories were AP2/ERF, MYB, NAC, WRKY, and bHLH (Fig. 4A). Depending on the ratio of number of DEG TFs to that of total TFs in one TF family, the top five TF family categories were WRKY, AP2/ERF, bZIP, NAC, and MYB (Fig. 4A). The expression patterns of some TF DEGs from seven families are presented in a heatmap (Fig. 4B). The expression patterns of most TFs can be classified into three types. The first one is that gene expression was induced after a short-term submergence (4 h, 12 h), but inhibited after a long-term submergence (24 h, 48 h), which comprised the major proportion of TF DEGs. The second is that after induction by short-term submergence, the high expression level persisted throughout the whole treatment period (e.g., EVM0007332 and EVM0050990). The third is that gene expression levels were higher before stress treatment, decreased after short-term submergence, and restored after long-term submergence (e.g., EVM0010523 and EVM0046641). Some TFs, such as EVM0047172 and EVM0000045 share similar expression patterns in submergence-tolerant and submergence-sensitive varieties, in which the expression levels were also the same. Other TFs shared similar expression patterns, but the expression levels were higher in the submergence-sensitive variety than that in the submergence-tolerant variety. Most TFs are members of the AP2/ERF family, such as EVM0043629, EVM0008900, and EVM0041629. A small portion of DEGs had different expression patterns in the submergence-tolerant and submergence-sensitive varieties (e.g., EVM0056506, EVM0029583, and EVM0049452). Fourteen members of the AP2/ERF GroupVII were identified from the Salix genome; their expression patterns are illustrated in Fig. 4C. According to their expression patterns, 14 genes can be classified into three categories. The expression levels of four genes on the top of the heatmap changed after submergence stress, with plants subjected to 24 h and 48 h treatment showing the highest expression levels. The second category, which included six genes, had higher expression levels at all treatment time points, with little variation between the treatments. The expression levels of the other four genes were very low (Fig. 4C).

Figure 4 Statistical analysis and heatmap illustration on TF DEGs.

(A) Statistical analysis of transcription factor (TF) differentially expressed genes (DEGs). (B) Heatmap illustration on the expression profiles of some TF DEGs from seven families. (C) Heatmap illustration of the expression profiles of 14 members from the AP2/ERF Group VII subfamily.

Identification on DEGs encoding for factors of carbohydrate metabolism, ROS signaling pathway, anaerobic respiration, and N-end rule pathway

Upon submergence stress, plants regulated the carbohydrate metabolism and ROS signaling pathway to cope with the hypoxia condition via the N-end rule pathway. We detected DEGs from these pathways, as illustrated in Figs. 5A and 5B. The ADH gene is a marker of fermentation; five Salix ADH genes with differential expression patterns were identified. Their expression levels reached a peak after 24 h treatment and were maintained at a higher level at 48 h, with the exception of EVM0046399_AtADH, whose expression level at 48 h decreased compared to the WYSL sample. It is worth noting that the expression levels of all five members were lower after 48 h in the WR samples than in the WYSL samples. SUS cleaves sucrose and provides monosaccharide substrates for glycolysis under submergence. Four SUS DEGs were found, and their expression was upregulated under longer submergence stress (24 h and 48 h). The expression levels of two members (EVM0011515_AtSUS1 and EVM0020822_AtSUS1) were higher in the WR samples than in the WYSL samples. Another three kinds of enzymes related to glycolysis were LDH, PDC, and AlaAT: 1 LDH, 4 PDC, and 4 AlaAT DEGs were found to be important factors in the fermentation pathway (Fig. 4A). Their expression levels were upregulated under hypoxia condition in both the WR and WYSL samples, with slight differences observed in several DEGs between WR and WRYL (e.g., EVM0013254_AlaAT and EVM0049446_AtPDC1).

Figure 5 Heatmap analysis of differentially expressed genes (DEGs) encoding for important players in submergence response.

(A) Heatmap analysis of differentially expressed genes (DEGs) encoding for important factors in carbohydrate metabolism and anaerobic respiration. (B) ROS signaling pathway and N-end rule pathway. (C) Ethylene, GA, and SA synthesis and signaling.

DEGs coding for enzymes involved in ROX production, signaling, and ROS detoxification are shown in a heatmap (Fig. 5B). In six AOX1A members, four members had a higher expression level after 4 h of treatment in the WRYL samples than in the WR samples. Six AOX1A members reached the maximum expression level after 4 h in the WRYL samples; however, this tendency was only observed for EVM0034773_AOX1A in the WR samples. While all five RBOHD DEGs had higher levels of expression in the WR samples, there were no evident differences in the levels of three kinds of antioxidant enzymes.

The expression profiles of PCO DEGs were inhibited after short-term submergence, and then upregulated under hypoxic conditions. DEG EVM0025544_PCO1 was found to have a higher level of expression in the WSYL sample at 24 h and 48 h (Fig. 5B).

Identification on DEGs encoding proteins in ethylene, GA, and salicylic acid (SA) synthesis and signaling

Five key genes in GA synthesis, GA20OXs, were identified. The expression levels of four of the five DEGs in WSYL were found to be higher than those in the WR samples. The GA receptor DEG EVM0004859_GID1C and GA-regulated protein GASA DEG were substantially upregulated in WSYL (Fig. 5C). For ethylene synthesis, five ACO genes were detected, and all five genes were found to be significantly upregulated at 4 h. Two genes, EVM0049609_ACO and EVM0029609_ACO, were upregulated in the WR samples, with the exception of EVM0047872_ACO. For SA signaling, SAR deficient 1 (SARD1) is a key regulator of ICS1 (isochorismate synthase 1) induction and SA synthesis. In the WR-CK sample, the expression levels of SARD1 were much higher than those in WSYL-CK. After 4 h of submergence, three genes were downregulated in WR but upregulated in WSYL. The ICS1 gene also showed an upregulated expression pattern in WR samples at 0, 4, and 12 h (Fig. 5C).

Hub genes were identified by co-expression network construction using WGCNA

A co-expression network using WGCNA was constructed to investigate the interrelationships among submergence responsive genes and identify the key regulators. According to the expression patterns of 10,097 DEGs (FPKM Value > 0.5), a hierarchical clustering tree (dynamic hybrid tree cut algorithm) was constructed and the DEGs were divided into 16 co-expression modules, with the highest number of DEGs (2,371) in the black module and the lowest number of DEGs (30) in the lightsteelblue1 module (Fig. S2, Table S13). The module trait correlation heatmap was drawn to show the correlation significance between the modules and submergence stress and willow samples with different hypoxia tolerance (Fig. S3). The modules with higher correlation significance (>0.7) and lower P-value (<0.05), the red and green modules, respectively, were selected for further analysis (Fig. S3). While the genes from red modules were induced rapidly after short-term submergence, higher expression levels did not last and reduced to a lower level during the rest of the submergence period. Another significant feature is that the expression levels of these genes were higher in the WYSL samples than in WR samples after 4 h submergence treatment (Fig. S4A). Genes from the green module also had significantly different expression profiles in the WYSL and WR samples. The peak of the expression level of these genes appeared in the WR samples without submergence stress, and then their expression levels decreased following submergence stress (Fig. S4B).

We performed functional enrichment analysis of genes in modules. The Clusters of Orthologous Groups (COG) function classification of green and red modules are presented in Fig. S5 and Table S14. The top three functional classes from the green module were G (carbohydrate transport and metabolism), T (signal transduction mechanisms), and Q (secondary metabolite biosynthesis, transport, and catabolism), while the top three classes from the red module were G, O (post-translational modification, protein turnover, chaperones), and T. Apart from the difference in top classes, the red and green modules had a similar enrichment pattern.

In the red and green modules, according to the kME value and correlation significance (Table S15), the hub genes were identified and the networks were constructed and visualized using Cytoscape 3.6.1. Eleven and five TFs were identified as hub genes for the red and green modules, respectively. In the red module, the 11 TFs included one GIF (EVM0054186), two trihelix TFs (EVM0004070, EVM0046362), three NAC TFs (EVM0056498, EVM0045598, EVM0025876), three WRKY TFs (EVM0008101, EVM0015670, EVM0006564), and two AP2/ERFs (EVM0004610, EVM0047172), among which GIF (EVM0054186), trihelix EVM0046362, and NAC EVM0056498 were located at the center of the network (Fig. 6A). In the green module, three MYBs and two NACs were illustrated at the center of the diagram (Fig. 6B). These TFs could be defined as candidate genes involved in the submergence stress response, and further functional evaluation of these genes will be carried out in the future.

Figure 6 Identification of hub genes in co-expression network under submergence stress.

(A) Eleven hub genes were identified from the red module. (B) Five hub genes were identified from the green module. The co-expression network was analyzed using WGCNA software and the graphic network was created by Cytoscape.

Verification of RNA sequencing results by qRT-PCR analysis of key genes

We selected 15 genes from the total number of DEGs identified for qRT-PCR to verify the reliability of RNA sequencing. The names and annotations of 15 genes, including putative hub TF genes, important factors in fermentation, hormone synthesis and signaling pathways, and oxygen level sensing are listed in Table S16. The expression patterns of these 15 genes are shown in Fig. 7. The qRT-PCR results showed that the expression profiles of all genes and the changes in expression after different periods of stress treatment were consistent with FPKM values (Table S9).

Figure 7 Verification of the DEGs with differentially expressed patterns under submergence stress by quantitative real-time PCR.

The gene expression profiles were evaluated using the 2−∆∆Ct method, and the control values were normalized to 1. Three biological replicates were performed for each sample. Bars represent the standard deviation of the mean.

Discussion

Submergence-tolerant varieties are capable of continuous growth under long-term hypoxia

According to a previous study, willows have one of the highest tolerances to flooding stress in trees (Mozo et al., 2021). However, the growth characteristics of willows during long-term submergence have not been reported previously. In our submergence experiments, different Salix varieties were found to have distinct growth capacities under long-term submergence stress. Submergence-tolerant varieties, such as “Suliu 795,” were able to maintain growth even after submergence lasting 2 months. When comparing the shoot length between 60 d and 66 d, the length of the shoots increased by 10 percent, suggesting that “Suliu 795” could cope with hypoxic conditions (Fig. 1). Comparative transcriptomic analyses of submergence-tolerant and submergence-sensitive varieties will help us uncover the mechanisms underlying these responses.

RNA sequencing results were reliable and indicated that large number of DEGs was involved in the submergence stress response

Two varieties with contrasting submergence tolerance abilities were used to perform a comparative transcriptomic analysis. Five samples were collected at the 0, 4, 8, 24, and 48 h submergence time points from one variety, and a total of ten samples were collected, where every sample had three replicates. The clean data from RNA sequencing had a high Q30 value and a high alignment ratio to the reference genome. PCA analysis showed that the three repeats from one sample clustered tightly together and in different samples separately (Fig. 2A). Verification of the expression of 15 genes by qRT-PCR showed that almost all genes had a similar expression oscillation pattern with the FPKM value obtained by high-throughput sequencing during the long-term submergence stress (Fig. 7). The results of these analyses indicated that the RNA sequencing data were reliable and could be used to conduct a thorough investigation of the molecular mechanisms of willow submergence. A large number of DEGs were identified between different time points after submergence stress in one variety (categories A, B, D, E) and also between the same time points from two varieties (category C) (Fig. 2B). The number of DEGs peaked after submergence for 24 h in both varieties, while the number of DEGs from two comparisons between 24 h and 48 h in the two varieties was the smallest. These data demonstrate that the regulation of gene expression peaked at 24 h in response to submergence stress, after which plants maintained their status in response to hypoxia. As a result, few DEGs were found in comparison between 24 h and 48 h (Fig. 2B). Shortly after submergence, the DEGs included more upregulated genes than downregulated genes. By contrast, after 12 h of submergence, the number of upregulated genes decreased and the number of downregulated genes increased. As a result, the DEGs included more downregulated genes than upregulated genes. These data indicate that after short-term submergence, plants began to attenuate the biological activities to adapt to the long-term submergence stress by downregulating the expression of some genes.

The Venn diagram signified the common and unique DEGs of the different comparisons (Fig. 2C). The Venn diagrams based on the comparisons between categories A, B, and C show that the portions of common DEGs for four or five comparisons were very small, with less than 10% of the total DEGs (Fig. 2C). The lower percentage of overlapped DEGs in categories A and B (5.19% and 9.16%, respectively) indicated that the gene expression changed markedly at different time points under submergence stress. Tolerant varieties mobilized more dynamic gene expression changes in response to submergence stress by decreasing the gene expression levels. The 5% common DEGs in category C implied that there were contrasting gene expression patterns as a result of the different willow varieties and different submergence treatment times (Fig. 2C).

GO analysis of DEGs revealed the molecular mechanisms underlying the submergence stress response

After submergence, the BP term “oxidation-reduction process (GO:0055114) ” was enriched in DEGs of all comparisons from categories A and B, indicating that from the beginning and through the 48 h submergence, the status of energy metabolism was always in dynamic change under submergence stress. At the beginning of the stress (4 h), the tolerant variety “Suliu 795” quickly initiated the braking action on the cellular activity through the enrichment of three GO terms containing “negative regulation (GO:0043086, GO:0010951, GO:1903507).” Thereafter, plants adjust their metabolic strategy by increasing the expression of genes from the categories “carbohydrate metabolic process (GO:0005975),” “nucleoside metabolic process (GO:0009116),” “response to nitrate (GO:0010167),” “nitrate transport (GO:0015706),” and “starch metabolic process (GO:0005982),” suggesting that plants begin the C and N destructive metabolism to provide the raw material for the anaerobic respiration under longer-term submergence stress. After submergence for 24 h, the term “anaerobic respiration (GO:0009061)” was enriched in the upregulated DEGs of WR-CK/WR-24h, indicating that tolerant plants convert aerobic metabolism to anaerobic metabolism successfully under hypoxia stress. To adapt to a long-term hypoxic environment, plants survive by generating oxygen through photosynthesis, with photosynthesis-related terms being found to be enriched in DEGs from WR-CK/WR-48h. From the enriched GO terms of DEGs from categories A and B, some identical terms (labeled blue) were added in the corresponding comparison from categories A and B, suggesting both varieties had a similar mechanism in the response to submergence stress (Fig. 3). Significant differences in enriched terms were also revealed (labeled red) (Fig. 3). The expression of the term “regulation of transcription, DNA-templated (GO:0006355),” which was enriched in the upregulated DEGs of all comparisons from category B, was found to be increased in many TFs after stress. The terms “hydrogen peroxide catabolic process (GO:0042744)” and “cellular oxidant detoxification (GO:0098869)” enriched in the upregulated DEGs of WYSL-CK/WYSL-48h also showed that sensitive plants did not establish the metabolic balance and energy saving mechanism after 48 h of submergence.

TFs played important roles in the submergence stress response

A previous study found that four classes of TFs, including AP2/ERF, bHLH, WRKY, and MYB, played significant roles in the flooding response circuitry of angiosperms (Reynoso et al., 2019). In the present study, 40% (1,844 out of 4,584) of annotated TF genes were DEGs, consistent with previously reported results. AP2/ERF, bHLH, WRKY, and MYB were the four top TF family categories. Differential expression patterns of DEG TFs implied their roles in routine regulation patterns in response to submergence stress, as well as in differential regulation in different varieties, resulting in differences in the tolerance ability of plants. The roles of ERFVII members in response to waterlogging stress have been extensively studied in rice and Arabidopsis (Fukao & Bailey-Serres, 2008; Barding et al., 2012; Niroula et al., 2012; Gibbs et al., 2011; Licausi et al., 2011). The expression patterns of 14 SmERFVII were detected, and only four members had upregulated expression profiles under submergence stress, as well as a higher FPKM value. However, similar expression patterns in WR and WYSL willows suggest that they were not the key regulators in increasing tolerance to submergence stress in submergence-tolerant willows. Several hub TF genes were identified after WGCNA analysis, with hub TFs from green and red modules mostly belonging to the MYB, AP2/ERF, WRKY, NAC, and Trihelix TF families. These TF genes possessed different expression profiles in the WR and WYSL willows, which may provide a molecular basis for differences in the tolerance of WR and WYSL. Although Trihelix TF, such as hypoxia response attenuator 1 (HRA1) does not belong to the five top TF family members, it is also a crucial component in hypoxia status sensing and cell activity regulation network. Further exploration of these TFs will uncover the mechanisms underlying the differences in the tolerance activities of different willows.

Submergence-tolerant willow “Suliu 795” may gain a higher tolerance via unique mechanisms

From the annotation of DEGs and the specific expression patterns of different modules, we found that WR plants may gain a higher submergence tolerance through several mechanisms (Fig. 8). First, the expression profiles of genes from the green and red modules were specific to WR and WYSL, respectively. In the green module, the group of genes was more highly expressed in the WR-CK sample; in the red module, the other group of genes was only substantially upregulated in WYSL-4h. The two group genes shared two COG function classification terms, “carbohydrate transport and metabolism” and “signal transduction mechanisms.” This implies that during the hydroponics culture stage in WR, hydroponics may mimic waterlogging stress. WR plants were very sensitive to low oxygen stress and initiated the appropriate mechanisms in advance to prepare for subsequent submergence stress.

Figure 8 The potential mechanisms for Submergence-tolerant willow “Suliu 795” gaining a higher tolerance.

Second, under longer submergence stress, the fermentation pathway was important for plants to survive, and the expression level of key genes in the fermentation pathway was indispensable. Five ADH genes, marker genes of fermentation, had a lower expression level at 48 h in the WR samples than in the WYSL samples, especially EVM0046399, whose expression declined sharply at 48 h compared with 24 h. Similar changes in gene expression were observed in the EVM0013254_AlaAT and EVM0049446_AtPDC1 genes. In some hypoxia-tolerant genotypes, the genes from the glycolysis pathway must be downregulated after long-term stress to maintain a continuous glycolytic flux to ensure the energy pool needed for the processes involved in the survival of trees under hypoxia (Salvatierra et al., 2020). The regulation of gene expression in WR plants is required to maintain energy homeostasis, which is essential to avoid the detrimental effects of energy depletion under hypoxic stress (Salvatierra et al., 2020).

Third, the genes responsible for ethylene, GA, and SA signaling were also found to have different expression patterns in the WR and WYSL plants. Ethylene acts as the primary signal for the majority of adaptations to flooding (Loreti, van Veen & Perata, 2016). The ethylene synthesis genes, EVM0047872_ACO and EVM0029609_ACO, increased in expression earlier in WR, at the 0 h and 4 h time points. By contrast, GA synthesis and signaling were enhanced in WYSL but reduced in WR, while SA and ethylene hormone synthesis were increased in WR. Willow plants may carry out the “quiescence” (low-O2 quiescence syndrome, LOQS) strategy, by responding quickly to submergence, with WR plants synthesizing ethylene quickly and inhibiting the GA product to suppress growth and maintain energy homeostasis to survive under conditions of long-term submergence. In addition to saving energy, WR plants may also create new sources of energy by obtaining oxygen via enhancing the expression of genes associated with photosynthesis and establishing the photosynthesis system in their roots.

Conclusion

In this study, the expression of a large number of genes was found to undergo significant changes under conditions of flooding stress, with a small proportion of shared genes between submergence-tolerant and submergence-sensitive varieties. Genes were found to be enriched in several categories of GO terms, including “oxidation-reduction process (GO:0055114),” “secondary metabolite biosynthetic process (GO:0044550),” “regulation of transcription, DNA-templated (GO:0006355)”, and “anaerobic respiration (GO:0009061)”. A total of 1,844 TF genes were identified as DEGs, with most belonging to the TF families of WRKY, AP2/ERF, bZIP, NAC, and MYB. Three different types of expression profiles of TFs provided a basis for the molecular mechanisms underlying the regulation of responses to flooding stress in willows, highlighting the different submergence tolerance abilities for the different varieties. WGCNA analysis revealed that several members of these families acted as hub genes in the regulation of the response to submergence stress. In addition to the differential expression pattern of TFs in the submergence-tolerant and submergence-sensitive varieties, the submergence-tolerant varieties may use three additional strategies to exert higher submergence tolerant activities. The first one is the hypersensitivity to low oxygen levels and a quick response to submergence stress; the second one is a steady supply of carbohydrates by slowing down the anabolic processes under hypoxic stress; the third one is keeping “quiescence” through fine-tuning the equilibrium between phytohormones including GA, SA and ethylene. Fourthermore, these plants create new sources of energy and obtain oxygen by enhancing the photosynthesis system gene expression and establishing a photosynthesis system in their roots (Fig. 8). Moreover, further functional analyses on candidate hub genes we identified in this study will help to reveal the mechanism of submergence tolerance regulation in Salix matsudana.

Supplemental Information

Supplemental Information 1 List and fpkm values of 22790 DEGs from 25 comparisons.

Click here for additional data file.

Supplemental Information 2 Supplementary Figures.

Click here for additional data file.

Supplemental Information 3 Supplemental Tables.

Click here for additional data file.

Supplemental Information 4 DEG sequences File 1.

Click here for additional data file.

Supplemental Information 5 DEG sequences File 2.

Click here for additional data file.

We thank Liwen Bianji, Edanz Editing China for editing the English text of a draft of this manuscript.

Additional Information and Declarations

Competing Interests

Author Contributions

Data Availability

The authors declare that they have no competing interests.

Yanhong Chen conceived and designed the experiments, performed the experiments, analyzed the data, prepared figures and/or tables, authored or reviewed drafts of the paper, and approved the final draft.

Jie Yang performed the experiments, prepared figures and/or tables, authored or reviewed drafts of the paper, and approved the final draft.

Hongyi Guo analyzed the data, authored or reviewed drafts of the paper, and approved the final draft.

Yawen Du performed the experiments, prepared figures and/or tables, and approved the final draft.

Guoyuan Liu performed the experiments, analyzed the data, prepared figures and/or tables, authored or reviewed drafts of the paper, and approved the final draft.

Chunmei Yu performed the experiments, analyzed the data, authored or reviewed drafts of the paper, and approved the final draft.

Fei Zhong performed the experiments, analyzed the data, authored or reviewed drafts of the paper, and approved the final draft.

Bolin Lian performed the experiments, analyzed the data, prepared figures and/or tables, and approved the final draft.

Jian Zhang conceived and designed the experiments, performed the experiments, authored or reviewed drafts of the paper, and approved the final draft.

The following information was supplied regarding data availability:

The S. matsudana genome sequences are available at NCBI: PRJNA687297.

The Raw RNA sequencing data is available at the CNGB Sequence Archive (CNSA) of China National GeneBank DataBase (CNGBdb): CNP0002062.

The assembled sequences of transcriptome (TSA) are available at the CNGB Sequence Archive (CNSA) of China National GeneBank DataBase (CNGBdb): CNA0038268.

The raw data for growth indicators (Fig. 1) are available in Table S2.

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
