# Peer review of "Comparative transcriptomic analysis reveals potential mechanisms for high tolerance to submergence in arbor willows"

_PeerJ, doi:10.7717/peerj.12881_

## Round 0.1 · original submission · Major Revisions

Two reviewers have made valuable comments on your manuscript, Please take comments from two reviewers into consideration when you make revisions.

·

Basic reporting

no comment

Experimental design

no comment

Validity of the findings

no comment

Additional comments

In the manuscript (64181-v0), Chen and his/her colleagues investigated the response to submergence at the genetic level using transcriptome analysis in arbor willows. The manuscript is useful for plant biologists aimed at breeding with improved tolerance to submergence stress. The purpose and the significance are sufficient. However, I think that this manuscript needs some revisions before publication.
Major:
1. The main shortcoming of the manuscript is that the writing skills of the manuscript are not very good. Some sentences are too long to get across. Overall, the manuscript is not well written. I suggest that the authors can turn to an English language editing company or a native English speaker.
2. The authors give three reasons for high tolerance of ‘SUliu 795’ to submergence. I think the conclusion is not valid and convincing, since physiological indicators were not determined.
3. GA metabolism and signaling is regulated by negative feedback regulation mechanism. As a result, the statement that reduced GA biosynthesis and signaling in tolerant variety under submergence is not appropriate.
4. The authors use actin as the reference gene to normalize gene expression in this work; however, the accuracy of the gene is not validated. Can you give some experimental data or some references to support this? It is recognized that selection of reference genes in specific condition is necessary.

Minor:
1. The title should be changed, approaches? reveals potential mechanisms for high tolerance to submergence in arbor willow?
2. Line 29: obtained is not appropriate here.
3. Line 112-113: the sentence should be rewritten.
4. Line 145: delete “has”
5. Line 149: change “in response to” to “exposed to”
6. Line 237-238: the sentence is too hard to get across.
7. Line 419: more upregulated?

Reviewer 2 ·

Basic reporting

The authors screened germplasm resources of arbor willows and discovered both submergence-tolerant and -sensitive varieties. Then, by performing RNA-seq, they compared the differences between the transcriptomes of two varieties and the different submergence treatment time points to identify the potential mechanisms of submergence in Salix and the unique approaches. Their findings provided valuable insights into the molecular mechanisms underlying the high tolerance to submergence of arbor willows . But looking at the full text, the article still has some problems.

Experimental design

The screening of the material is only through the four phenotypic indicators, is it reliable? Is there a physiological indicator to assist?

Validity of the findings

The purpose of this study is to reveal the pathway and molecular mechanism of the high flooding tolerance of arbour willows. However, Figure 8 only describes the hypothetical model of "Suliu 795" gaining high flooding tolerance. The author needs to add a picture or improve this picture.

Additional comments

References are cited too frequently in the Introduction, can some statements extract overview? For example, it refers to flooding is the main threat to plant growth and survival, but there is no universal impact of water flooding on plants.
What is the difference between the heat maps and common thermostats present in Figure 4, and C is not as intuitive in conventional thermostats.
The conclusion part of the article only summarizes the results of the study, whether the author needs to join the author's future prospects of this study.
The article has screened germplasm resources of arbor willows and discovered both submergence-tolerant and -sensitive varieties., why not compare the difference of the two materials' response to flooding with a graph, so that it will be more intuitive.

---

## Round 0.2 · Major Revisions

Before this manuscript can be accepted, please address the following comments from the Section Editor:

Gerard Lazo, the Section Editor, has commented and said:

> The 22790 DEG should be deposited in a third-party resource such as NCBI.
>
> The supplemental data is provided in RAR format; this is not a common format used and there is incompatibilities associated with different versions of this software. I would recommend using TAR, or compression with ZIP or GZIP. At this point I am unable to extract data from the furnished RAR file.
>
> The manuscript appears in need of major revision as there is no connection to sequence data and cited annotations which have been described. In addition, the annotations should have coded values also available (e.g. GO:12345). These can be provided in tables as the general summary figures are just that, general summaries, but with no reference points to work from to DEGs which appear to be an important part of the manuscript. Likewise the Venn diagrams are also summaries and do not provide links to the DEGs for critical analysis. Tables need to be generated which can make such descriptions easily resolvable. Meaning lots of pretty pictures which have no navigational consequence.
>
> Journal manuscripts are often scanned by text-mining software that locates and extracts core data elements, like gene function. Adding standard ontology terms, such as the Gene Ontology (GO, geneontology.org) or others from the OBO foundry (obofoundry.org) can enhance the recognition of your contribution and description. This will also make human curation of literature easier and more accurate. None of this was visible.

·

Basic reporting

no comment

Experimental design

no comment

Validity of the findings

no comment

Additional comments

The authors have solved most of the issues I suggested.

---

## Round 0.3 · Minor Revisions

Please respond to the Section Editor's comments:

"The authors went through great strides to update the information as requested; however, I am still at a loss to find the actual sequences that correspond to the 22790 DEGs discussed. Are the EVM000… sequences housed in a common repository somewhere? I failed to find them anywhere. There is a lot of discussion and binning of these sequences, but without any sequence to work with it means nothing. In other regards the tables and figures are helpful; however, the only thing that will connect everything is the sequence data for which everything is described. I will approve or agree with the acceptance once this is done. Am I missing a clear statement somewhere about where the sequences are – I did not see it nor was able to find it.

Reviewer 2 ·

Basic reporting

no comment

Experimental design

no comment

Validity of the findings

no comment

Additional comments

It is suggested to further verify the key genes found, which will significantly improve the paper in terms of verification function and interaction, so as to have a clearer understanding of the response mechanisms for high tolerance to submergence in arbor willows.

---

## Round 0.4 · Minor Revisions

The Section Editor has commented and said:

"The sequences have been provided, so the data is there. However, the assembled sequence data should be placed in a third-party repository. Assembled transcriptomes can be deposited as a transcriptome shotgun assembly (GenBank TSA resource). Please see: https://www.ncbi.nlm.nih.gov/nuccore/.

A few suggested edits are listed below.

LINE NO: / BEFORE / AFTER / [COMMENTS]
LINE 38: / hypoxic stress; (3) / hypoxic stress; and (3) / [.]
LINE 77: / Gibberellins Aacid / Gibberillic acids / [or gibberillins?]
LINE 84: / . / . / [ remove period(.) before citation ]
LINE 113: / These reseaeched / This research / [.]
LINE 113: / . / . / [ remove period(.) before citation ]
LINE 119: / . / . / [ remove period(.) before citation ]
LINE 122: / plants mainly / plants were mainly / [.]
LINE 123: / such as Chrysanthemum / such as with Chrysanthemum / [.]
LINE 132: / Arabidopsis.. / Arabidopsis. / [one period(.)]
LINE 193: / . / . / [A reference genome version number should be provided as reference genomes are often updated.]
"

---

## Round 0.5 · Minor Revisions

Please incorporate the sequence number information into text, or add data available at end of text: The S. matsudana genome sequences are available at NCBI: PRJNA687297; the Raw RNA sequencing data is available at the CNGB Sequence Archive (CNSA) of China National GeneBank DataBase (CNGBdb): CNP0002062. The Assembled sequences of transcriptome (TSA) is available at the CNGB Sequence Archive (CNSA) of China National GeneBank DataBase (CNGBdb): CNA0038268

---

## Round 0.6 · accepted · Accept

The authors made the change according to the comments. The manuscript is acceptable.